# Incidence, predictors, and outcomes of DAPT non-compliance in planned vs. ad hoc PCI in chronic coronary syndrome

**Jahanzeb Malik**[1]*, **Husnain Yousaf**[2], **Waleed Abbasi**[1], **Nouman Hameed**[2], **Muhammad Mohsin**[1], **Abdul Wahab Shahid**[1], **Mahnoor Fatima**[1]

**1** Rawalpindi Institute of Cardiology, Rawalpindi, Pakistan, **2** Armed Forces Institute of Cardiology, Rawalpindi, Pakistan

* Heartdoc86@gmail.com

**Data Availability Statement:** All relevant data are within the manuscript and its Supporting Information files.

## Abstract

### Objective

The disruption of dual antiplatelet therapy (DAPT) causes more adverse events after percutaneous coronary intervention (PCI). However, incidence and predictors of DAPT non-compliance are unknown in chronic coronary syndrome patients when compared between planned and ad hoc PCI.

### Methods

This investigation was aimed to assess the incidence, predictors, outcomes, and primary mode of non-compliance of DAPT in patients with chronic coronary syndrome undergoing their first PCI. We analyzed the patients between planned (group 1) and ad hoc (group 2) PCI.

### Results

There were a total of 628 participants in this investigation (370 were in planned PCI and 270 in the ad hoc PCI group). Out of 628 patients, by one month, 10% left DAPT in planned PCI group and 19.7% in ad hoc PCI group (aOR: 0.451, 95% CI: 0.285–0.713, p = 0.001). At 12 months, DAPT non-compliance was significantly more in ad hoc PCI group (52.7% vs. 47.8%; aOR: 0.647 95% CI: 0.470–0.891, p = 0.008). Age > 65 years (p < 0.001), low education status (p = 0.012), residents of rural areas (p < 0.001), ad hoc PCI group (p = 0.036), and angina class II (p = 0.038) were predictors for DAPT non-compliance in this cohort.

### Conclusion

Approximately 5 out of 10 patients disrupt DAPT due to non-compliance. This investigation provides an insight on additional predictors of non-compliance to DAPT, helping us to identify and address specific patient-related factors for disruption.

**Funding:** The author(s) received no specific funding for this work.

**Competing interests:** The authors have declared that no competing interests exist.

## Introduction

Ad hoc percutaneous coronary intervention (PCI), defined as performing PCI following a diagnostic coronary angiogram, is becoming a common practice in catheterization labs worldwide, for several reasons. The most important advantage is the patient preference to avoid multiple invasive procedures, and cost-effectiveness when performed without major complications [1]. With improved safety profiles of catheterization procedures, the guidelines for ad hoc PCI have been added to Cardiovascular Angiography and Interventions (SCAI) consensus statement in 2013, and the European Society of Cardiology (ESC) in 2018 [2, 3]. However, both guidelines rely on institutional protocols for ad hoc PCI and no patient-specific criteria are available. Several observational studies have shown similar clinical outcomes after planned PCI and ad hoc PCI but clinical evaluation of chronic coronary syndrome has been limited because of the inclusion of acute coronary syndrome (ACS) patients along with chronic coronary syndrome cohort [4, 5]. Furthermore, no studies have compared compliance to dual antiplatelet therapy (DAPT) between planned and ad hoc PCI in chronic coronary syndrome.

According to a report from World Health Organization (WHO), approximately 50% of the patients are non-adherent to prescribed medications, leading to increased hospital visits and disease progression [6]. Besides, a WHO medication adherence report exhibited an increased cost of approximately $100 billion per annum due to poor compliance to medicines [7]. Medicine non-compliance is observed commonly in patients with cardiovascular disease, in part due to alleviation of symptoms after revascularization [8]. In patients undergoing PCI, adherence to DAPT is vital in the prevention of thrombotic complications and the use of aspirin in addition to a P2Y12 inhibitor is necessary to prevent morbidity and mortality after stent implantation [9]. Some reports suggest high rates of medicine non-adherence post-procedure where ample time and counseling has not been provided beforehand. Therefore, we conducted this study to specify incidence, predictors, and 12-month outcomes on DAPT non-adherence in a cohort of planned PCI compared with ad hoc PCI patients.

## Methods

### Study design

This observational study was approved by the ethical review board of Armed Forces Institute of Cardiology (ID#AFIC/19/IRB/23). Written, informed consent was taken from the participants before data collection. Data were collected according to World Medical Declaration of Helsinki between February 2019 to December 2020. The primary objective of our study was to assess the incidence and predictors of non-adherence (either brief or permanent cessation) to DAPT in patients with chronic coronary syndrome undergoing their first PCI. There were two groups defined in this study. Those who underwent planned and ad hoc PCI were labeled group 1 and 2, respectively.

### Patient selection criteria

All patients, irrespective of age and gender, undergoing first PCI for chronic coronary syndrome, despite optimal medical treatment were enrolled in the study. After coronary angiogram, the Heart team decided on a deferred (planned) approach (group 1) or an ad hoc approach (group 2) for PCI. To eliminate bias, the decision was not made by any of the authors. All PCIs were done by interventional cardiologists at our institute (WA, KS, JM). Patients with prior PCI or coronary artery bypass graft (CABG), ischemic/non-ischemic cardiomyopathies, end-stage renal disease, atrial fibrillation (on oral anticoagulants) and acute coronary syndrome during the last three months were excluded from the study. Patients with

allergy to aspirin and clopidogrel and discontinuation on physician orders or due to any bleeding complications were also excluded. All medicine was reimbursed through the government channel and no cost levied on the patient to eliminate the attrition bias. DAPT were prescribed as per our institutional protocols post-PCI (for a period of twelve months regardless of ACS or CCS. DAPT is continued for more than twelve months in patients with PRESICE-DAPT score < 25).

## Definitions

Chronic coronary syndrome suitable for PCI was defined as per ESC guidelines 2019 [3]: (1) In patients with mild symptoms (Canadian Cardiovascular Society angina class I) or no symptoms receiving optimal medical therapy, in which non-invasive test showed high-risk features were selected for revascularization for improvement of prognosis after invasive coronary angiogram showing $\geq$ 70% stenosis in an epicardial coronary artery (2) In symptomatic patients (Canadian Cardiovascular Society angina class II or more), in whom invasive coronary angiogram showed $\geq$ 70% stenosis in an epicardial coronary artery. DAPT non-adherence was defined as cessation of antiplatelet medication without any side effects or physician order. Reinstitution within 30 days was called brief cessation and > 30 days' cessation was labeled as permanent cessation. This classification was not overlapped during the course of the study. Major adverse cardiovascular events (MACE) were defined as the composite of all-cause death, cardiac death, target vessel revascularization (TVR), and myocardial infarction (MI) (stent thrombosis or otherwise) according to the ARC criteria [10]. Death was classified as a cardiac or non-cardiovascular cause. TVR was defined as repeat intervention or CABG of the target vessel. MI was defined as symptoms and electrocardiographic changes consistent with myocardial ischemia along with rise of cardiac biomarkers with at least one value above the 99th percentile of the upper border limit.

## Follow-up

Follow-up was on the telephone conducted by the authors at 30 days, and 12 months. In case of DAPT non-compliance, detailed information about the drug name, dates of stopping and restarting, and reasons for stopping were obtained.

## Statistical analysis

Statistical analysis was performed on Statistical Package for the Social Sciences (SPSS) version 26 (IBM Corp, Armonk, NY, USA). Continuous variables were presented as mean ± standard deviation (SD) and were tested for normal distribution by the Shapiro-Wilk test. Normally distributed variables were compared using Student's t-test and non-normally distributed variables were compared with the Mann-Whitney U test. Categorical variables, presented as frequency and percentages, were compared using the Chi-square test. Multivariate logistic regression was used for predictors of non-compliance to DAPT. Odds ratio (OR) and confidence interval (CI) were presented for non-compliance to DAPT. Cox regression model was used for DAPT non-compliance between planned PCI and ad hoc PCI groups at 12 months. P-value <0.05 was taken as significant.

## Results

Of 628 patients enrolled in our study, 370 were in planned PCI and 270 in the ad hoc PCI group. Regarding the baseline characteristics, the ad hoc PCI had slightly, but significantly, lower prevalence of old age, education level, and higher prevalence of hypertension (HTN).

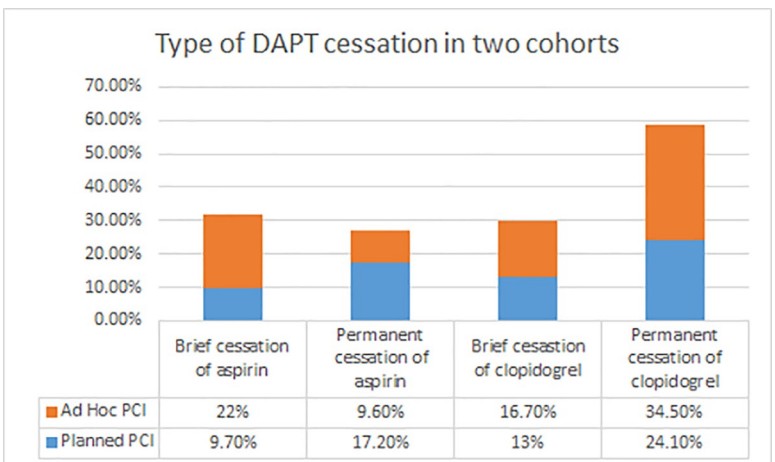

**Fig 1. Brief cessation (<30days) and permanent cessation (>30days) of DAPT in planned PCI and ad hoc PCI cohorts.** All four groups have p < 0.05.

More than half of the patients were diabetics or smokers and had abnormal lipid levels. Approximately one-third had prior MI while peripheral arterial disease was less common in the two cohorts. Compared with patients in the planned PCI group, brief non-adherence with aspirin and clopidogrel was more common in the ad hoc PCI group (p < 0.001, 0.002, respectively). Permanent non-adherence of aspirin was more in planned PCI while clopidogrel was more common in the ad hoc PCI group (Fig 1). Baseline characteristics and compliance to DAPT are exhibited in Table 1.

In procedural characteristics, left main stem (LMS) and multi-vessel procedures were more common in planned PCI while single vessel was treated more in the ad hoc PCI group. All-cause mortality was insignificant between the two groups but cardiovascular death was increased in the ad hoc PCI group due to non-compliance to DAPT (p < 0.001). Table 2 summarizes procedural characteristics and MACE in both groups.

The cumulative incidence of DAPT non-compliance and individual drug non-compliance is shown in Table 3. By one month, 10% left DAPT in planned PCI group and 19.7% in ad hoc PCI group (aOR: 0.451, 95% CI: 0.285–0.713, p = 0.001). At 12 months, DAPT non-compliance was significantly more in ad hoc PCI group as compared to planned PCI group (52.7% vs. 47.8%; aOR: 0.647 95% CI: 0.470–0.891, p = 0.008) (Fig 2). Age > 65 years (p < 0.001), low education status (p = 0.012), residents of rural areas (p < 0.001), ad hoc PCI group (p = 0.036), and angina class II (p = 0.038) were predictors for DAPT non-compliance in this cohort (Table 4). Fig 2 demonstrates the Cox regression model for DAPT non-compliance after 12 months' follow-up.

## Discussion

The key findings of our study are as follows: 52.7% of the patients who had ad hoc PCI disrupted DAPT due to non-compliance when compared with planned PCI (41.8%). Patients non-compliant to DAPT tended to be older, had angina CCS class II, lived in rural areas, were less educated, and had undergone ad hoc PCI. DAPT non-compliance led to an increased cardiovascular death in ad hoc PCI whereas all-cause death was associated with planned PCI. Left main, and multivessel PCI was more common in the planned PCI group.

To our knowledge, the present study is the first to compare the characteristics and outcomes of patients with DAPT non-compliance between ad hoc and planned PCI cohorts.

**Table 1. Baseline characteristics and compliance in both planned and ad hoc PCI group.**

| Variable | Total (n = 628) | Planned PCI (n = 370) | Ad Hoc PCI (n = 258) | p-value |
|---|---|---|---|---|
| **Age (years) (mean ± SD)** | 56.55 ± 14.24 | 57.52 ± 14.97 | 55.16 ± 13.01 | **0.041**[*] |
| **Gender n(%)** | | | | 0.143 |
| Males | 455 (72.5%) | 260 (70.3%) | 195 (75.6%) | |
| Females | 173 (27.5%) | 110 (29.7%) | 63 (24.4%) | |
| **BMI (kg/m$^2$)** | 27.42 ± 3.53 | 27.47 ± 3.54 | 27.35 ± 3.52 | 0.671 |
| **Dyslipidemia n(%)** | 340 (54.1%) | 198 (53.5%) | 142 (55%) | 0.706 |
| **Smoking n(%)** | 408 (65%) | 246 (66.5%) | 162 (62.8%) | 0.340 |
| **PAD n(%)** | 85 (13.5%) | 53 (14.3%) | 32 (12.4%) | 0.489 |
| **Prior MI n(%)** | 204 (32.5%) | 120 (32.4%) | 84 (32.6%) | 0.974 |
| **Diabetes n(%)** | 352 (56.1%) | 203 (54.9%) | 149 (57.8%) | 0.473 |
| **Hypertension n(%)** | 393 (62.4%) | 217 (58.6%) | 175 (67.8%) | **0.019**[*] |
| **Angina Class n(%)** | | | | 0.850 |
| NYHA II | 264 (42%) | 156 (42.2%) | 108 (41.9%) | |
| NYHA III | 315 (50.2%) | 187 (50.5%) | 128 (49.6%) | |
| NYHA IV | 49 (7.8%) | 27 (7.3%) | 22 (8.5%) | |
| **Region n(%)** | | | | 0.922 |
| Urban | 332 (52.9%) | 195 (52.7%) | 137 (53.1%) | |
| Rural | 296 (47.1%) | 175 (47.3%) | 121 (46.9%) | |
| **Education level n(%)** | | | | **0.010**[*] |
| More than secondary school | 257 (40.9%) | 167 (45.1%) | 90 (34.9%) | |
| Less than secondary school | 371 (59.1%) | 203 (54.9%) | 168 (65.1%) | |
| **EF(%)** | 47.93 ± 11.54 | 48.03 ± 11.50 | 47.79 ± 11.62 | 0.801 |
| **Number of Medicines (mean ± SD)** | 6.93 ± 2.40 | 6.95 ± 2.40 | 6.90 ± 2.39 | 0.794 |
| **Single pill formula for DAPT n(%)** | 395 (62.9%) | 239 (64.6%) | 156 (60.5%) | 0.292 |
| **Poly pill formula for DAPT n(%)** | 233 (37.1%) | 131 (35.4%) | 102 (39.5%) | 0.314 |
| **Compliance to aspirin n(%)** | | | | |
| First month | 590 (93.9%) | 358 (96.8%) | 232 (89.9%) | **<0.001**[*] |
| Twelve months | 541 (86.1%) | 328 (88.9%) | 212 (82.2%) | **0.016**[*] |
| **Compliance to clopidogrel n(%)** | | | | |
| First month | 565 (90%) | 342 (92.4%) | 223 (86.4%) | **0.014**[*] |
| Twelve months | 396 (63.1%) | 250 (67.6%) | 146 (56.6%) | **0.005**[*] |
| **Aspirin non-adherence n(%)** | | | | |
| Brief (<30 days) | 93 (14.8%) | 36 (9.7%) | 57 (22.1%%) | **<0.001**[*] |
| Permanent (>30 days) | 89 (14.2%) | 64 (17.3%) | 25 (9.7%) | **0.018**[*] |
| **Clopidogrel non-adherence n(%)** | | | | |
| Brief (<30days) | 91 (14.5%) | 48 (13%) | 43 (16.7%) | **0.002**[*] |
| Permanent (>30 days) | 178 (28.3%) | 89 (24.1%) | 89 (34.5%) | **0.002**[*] |
| **Warfarin n(%)** | 148 (23.6%) | 78 (21.1%) | 70 (27.1%) | 0.079 |

Continuous variables presented as mean ± standard deviation (SD) and were compared using Student's t test for normal distribution and Mann-Whitney U test for abnormal distribution. Categorical variables presented as proportions n (%) and were compared using Chi-square test. Body mass index (BMI), peripheral arterial disease (PAD), myocardial infarction (MI), New York Heart Association (NYHA), ejection fraction (EF), dual-antiplatelet therapy (DAPT).

Consistent with previous studies, DAPT non-compliance was higher in old age and was associated with higher rates of MACE following stent placement [9, 11]. Two large registries have investigated non-adherence to DAPT. The PARIS registry (n = 5018) was a prospective observational study that examined different modes of DAPT cessation in patients undergoing PCI

**Table 2. Procedural characteristics and MACE.**

| Variable | Total (n = 628) | Planned PCI (n = 370) | Ad Hoc PCI (n = 258) | p-value |
|---|---|---|---|---|
| **PCI vessel** | | | | |
| LMS | 33 (5.3%) | 26 (7%) | 7 (2.7%) | **0.017*** |
| LAD | 376 (59.9%) | 226 (61.1%) | 150 (58.1%) | 0.459 |
| LCx | 245 (39%) | 149 (40.3%) | 96 (37.2%) | 0.439 |
| RCA | 260 (41.4%) | 153 (41.4%) | 107 (41.5%) | 0.976 |
| **Number of vessels involved** | | | | 0.120 |
| One | 401 (63.9%) | 225 (60.8%) | 176 (68.2%) | |
| Two | 173 (27.5%) | 108 (29.2%) | 65 (25.2%) | |
| Three | 54 (8.6%) | 37 (10%) | 17 (6.6%) | |
| **Number of vessels treated** | | | | **<0.001*** |
| One | 556 (88.5%) | 308 (83.2%) | 248 (96.1%) | |
| Two | 52 (8.3%) | 45 (12.2%) | 7 (2.7%) | |
| Three | 20 (3.2%) | 17 (4.6%) | 3 (1.2%) | |
| **Number of stents implanted** | | | | **0.012*** |
| One | 466 (74.2%) | 260 (70.3%) | 206 (79.8%) | |
| Two | 117 (18.6%) | 83 (22.4%) | 34 (13.2%) | |
| Three | 45 (7.2%) | 27 (7.3%) | 18 (7%) | |
| **Stent type** | | | | **0.002*** |
| BMS | 66 (10.5%) | 33 (8.9%) | 33 (12.8%) | |
| DES 1st generation | 166 (26.4%) | 83 (22.4%) | 83 (32.3%) | |
| DES 2nd generation | 396 (63.1%) | 254 (68.6%) | 142 (55%) | |
| **All-cause mortality** | 96 (15.3%) | 61 (16.5%) | 35 (13.6%) | 0.317 |
| Cardiovascular Death | 62 (9.9%) | 22 (5.9%) | 40 (15.5%) | **<0.001*** |
| MI | 45 (7.2%) | 27 (7.3%) | 18 (7%) | 0.878 |
| TVR | 53 (8.4%) | 33 (8.9%) | 20 (7.8%) | 0.605 |

Categorical variables presented as proportions n (%) and were compared using Chi-square test. Percutaneous coronary intervention (PCI), left main stem (LMS), left anterior descending (LAD), left circumflex (LCx), right coronary artery (RCA), bare metal stent (BMS), drug-eluting stent (DES), major adverse cardiovascular events (MACE), myocardial infarction (MI), target vessel revascularization (TVR).

due to coronary artery disease [9]. The EDUCATE registry (n = 2265) was another prospective observational study, which focused on non-adherence to DAPT [12]. Both had a non-adherence rate of 9.6% at one-year and six-month follow-up, respectively. In contrast to that, our cohort shows 46.3% overall non-compliance to DAPT at one-year follow-up and more non-

**Table 3. Non-compliance of DAPT in planned PCI and ad hoc PCI cohorts.**

| Variable | Planned PCI | Ad Hoc PCI | OR (95% CI) | P-value |
|---|---|---|---|---|
| **Non-compliance to aspirin at 1 month** | 12 (3.2%) | 26 (10%) | 0.322 (0.165–0.626) | **<0.001*** |
| **Non-compliance to aspirin at 12 months** | 41 (11%) | 46 (17.8%) | 0.622 (0.421–0.918) | **0.016*** |
| **Non-compliance to clopidogrel at 1 month** | 28 (7.5%) | 35 (13.5%) | 0.558 (0.348–0.893) | **0.014*** |
| **Non-compliance to clopidogrel at 12 months** | 120 (32.4%) | 112 (43.4%) | 0.747 (0.610–0.915) | **0.005*** |
| **Non-compliance to DAPT at 1 months** | 37 (10%) | 51 (19.7%) | 0.506 (0.342–0.749) | **0.001*** |
| **Non-compliance to DAPT at 12 months** | 155 (41.8%) | 136 (52.7%) | 0.795 (0.673–0.939) | **0.007*** |

Variables presented as n (%) with OR (95% CI).

*p-value < 0.05 taken as significant. Percutaneous coronary intervention (PCI), odds ratio (OR), confidence interval (CI), dual antiplatelet therapy (DAPT).

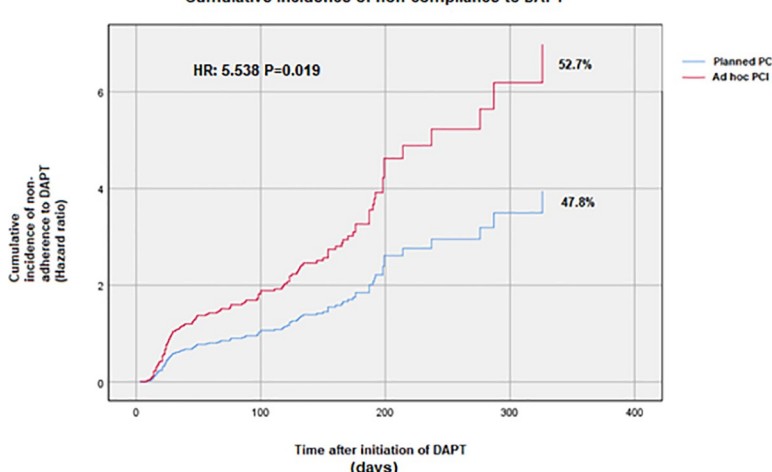

**Fig 2. Cumulative incidence of non-compliance to DAPT at 12 months' follow-up.** Dual antiplatelet therapy (DAPT), percutaneous coronary intervention (PCI).

compliance was associated with the ad hoc PCI. The independent predictors of non-compliance in PARIS and EDUCATE were bleeding, age, gender, prior coronary artery disease, and discharge medications while our study had predictors related to the level of education and living.

There has not been a randomized controlled trial comparing ad hoc PCI with planned PCI in patients with chronic coronary syndrome assessing the level of DAPT adherence between the two groups. Several previous studies are observational and non-randomized in nature. They found similar outcomes for both ad hoc and planned PCI [4, 5]. Regarding cardiovascular death, two studies show conflicting results of ad hoc PCI on long-term mortality. In New York PCI Reporting System, ad hoc PCI was associated with a lower risk of MACE at 36-months while IRIS-DES demonstrated neutral results of ad hoc vs. planned PCI [13, 14]. Our study population shows a positive relationship between cardiovascular death and the type of PCI strategy. However, there is no study investigating the DAPT compliance and associated factors in patients with chronic coronary syndrome because all previous studies have investigated either procedural outcomes or included a large number of ACS patients giving a rationale to perform ad hoc PCI as an integral part of acute MI.

Modern cardiology excels in treating life-threatening emergencies like acute MI due to the availability of contemporary methods of diagnosis and treatment. However, an untoward attitude on counseling regarding medicine adherence is observed in cardiology practice. In a

**Table 4. Logistic regression analysis to assess the predictors for non-compliance to DAPT.**

| Variable | B | S.E | Wald | OR (95% CI) | p-value |
|---|---|---|---|---|---|
| Age (>65 yrs.) | 0.038 | 0.010 | 14.438 | 1.039 (1.019–1.059) | <0.001 |
| Education level (less than secondary school) | -0.463 | 0.184 | 6.352 | 0.629 (0.439–0.902) | 0.012 |
| Region (Rural) | -1.414 | 0.184 | 59.351 | 0.243 (0.170–0.348) | <0.001 |
| Ad Hoc PCI | -0.385 | 0.184 | 4.376 | 0.680 (0.474–0.976) | 0.036 |
| Angina class II | -0.727 | 0.350 | 4.324 | 0.483 (0.244–0.959) | 0.038 |

Beta (B), standard error (SE), dual antiplatelet therapy (DAPT), percutaneous coronary intervention (PCI), odds ratio (OR), confidence interval (CI). p-value < 0.05 taken as significant.

report by WHO, medicine non-compliance is multifactorial, normally classified into five categories: socioeconomic background, intervention-related factors, patient belief, health condition-related circumstances, and health-care system [6]. Among the above-mentioned factors, patient-related factors are most important in cardiovascular disease because poor compliance can severely compromise disease outcomes and increase mortality rates. Lack of involvement in the treatment decision-making process or suboptimal health literacy can contribute to medication non-compliance. In the United States alone, an estimated 90 million people have deficient knowledge about their health, placing them at an increased risk of adverse events [15]. In our study, a trend towards ad hoc PCI, where a patient has little or no time in decision-making, can be a contributing factor towards DAPT non-compliance. Furthermore, studies of patients with cardiovascular disease have shown that anxiety and depression are common with coronary artery disease [16, 17]. This was not assessed in the present study, but it can present as an important factor for DAPT non-compliance [18].

## Limitations

The present investigation was an observational study that ruled out contributive results, possibly introducing the possibility of confounding on risk estimates. Medication use was self-reported by patients, which can introduce misclassification bias and the possibility of recall bias. More pertinent methods of quantifying compliance such as pill count or metabolite screening were not employed in this investigation. Platelet aggregation analysis was not performed, which can lead to underreporting of non-compliance. Furthermore, data regarding non-DAPT drug adherence was not collected and patient genotyping for gene metabolites was not performed, which could have affected the efficacy of DAPT and influence MACE outcomes.

## Conclusions

Approximately 5 out of 10 patients disrupt DAPT due to non-compliance. The incidence of non-compliance increases in ad hoc PCI at 12 months and clopidogrel is the most commonly missed antiplatelet in our cohort. DAPT non-compliance resulted in a higher incidence of cardiovascular deaths. This data provides an insight on additional predictors of non-compliance to DAPT, helping us to identify and address specific patient-related factors for disruption, and put the patients at lower risk for MACE and mortality with DAPT compliance. As suggested by this investigation, a greater effort should be made to educate patients for prescribing dual antiplatelet therapy, and the risks of medication non-compliance.

## Supporting information

**S1 File.**
(SAV)

## Author Contributions

**Conceptualization:** Jahanzeb Malik.

**Data curation:** Jahanzeb Malik, Waleed Abbasi, Muhammad Mohsin, Abdul Wahab Shahid.

**Formal analysis:** Jahanzeb Malik, Waleed Abbasi, Mahnoor Fatima.

**Methodology:** Jahanzeb Malik, Husnain Yousaf, Mahnoor Fatima.

**Supervision:** Muhammad Mohsin.

**Writing – original draft:** Jahanzeb Malik, Husnain Yousaf, Nouman Hameed, Abdul Wahab Shahid.

**Writing – review & editing:** Jahanzeb Malik, Waleed Abbasi, Nouman Hameed, Muhammad Mohsin.

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
