## [Decision Letter · Decision Letter 0]

7 Jun 2021

PONE-D-21-12701

Incidence, Predictors, and Outcomes of DAPT Non-compliance in Planned vs. Ad Hoc PCI in Chronic Coronary Syndrome

PLOS ONE

Dear Dr. Malik,

Thank you for submitting your manuscript to PLOS ONE. After careful consideration, we feel that it has merit but does not fully meet PLOS ONE’s publication criteria as it currently stands. Therefore, we invite you to submit a revised version of the manuscript that addresses the points raised during the review process.

The reviewers generally found the manuscript and revisions acceptable, but still have considerable questions regarding multiple parts of the manuscript. Please pay careful attention these issues in an itemized fashion, and we look forward to your revisions. 

We look forward to receiving your revised manuscript.

Kind regards,

R. Jay Widmer

Academic Editor

PLOS ONE

Journal Requirements:

Reviewers' comments:

Reviewer's Responses to Questions

**Comments to the Author**

1. Is the manuscript technically sound, and do the data support the conclusions?

Reviewer #1: Yes

2. Has the statistical analysis been performed appropriately and rigorously? 

Reviewer #1: Yes

3. Have the authors made all data underlying the findings in their manuscript fully available?

Reviewer #1: Yes

4. Is the manuscript presented in an intelligible fashion and written in standard English?

Reviewer #1: Yes

5. Review Comments to the Author

Reviewer #1: Dear Authors,

thank you for your valuable contribution and congratulation on your work in this not well-defined scenario. The paper is well-written considering English fluency, and the study is well-conducted about major biases.

I have some comments:

The abstract resulted in quite a few data. I would suggest making it more informative adding the number of patients for each group and the statistically significant data about the predictors.

Which therapy as DAPT was indicated? Aspirin and Clopidogrel in all patients? And the length of the therapy was previously established according to some risk score, bleeding risk of the patients or the complexity of the PCI?

Please specify.

“DAPT non-adherence was defined as cessation of antiplatelet medication without any side effects or physician order.” What in case of side effect such as thrombosis? How do you consider these patients?

The Target Lesion Revascularization is not considered. Can you explain the reason for this choice?

The last sentence of the Statistical analysis paragraph misses the final period.

Please add the statistical value in the Results for the predictors.

Please specify the abbreviations of the statistical value of the Table 4 in its caption (e.g. B, S.E.)

You showed only the predictors with statistically significant value, but you discussed the predictors of the PARIS and EDUCATED trial. What about the impact of the predictors of these trials in your population? Please, it could be relevant if you have the chance to produce the statistical analysis of these predictors and discuss them.

Thank you very much for your attention and for your interesting work.

6. PLOS authors have the option to publish the peer review history of their article (what does this mean?). If published, this will include your full peer review and any attached files.

Reviewer #1: **Yes: **Alessandro Sticchi, MD.

---

## [Author Response · Author response to Decision Letter 0]

8 Jun 2021

Dear Sir 

Thank you for reviewing our manuscript. Your valuable comments made this better.

Question 1

The abstract resulted in quite a few data. I would suggest making it more informative adding the number of patients for each group and the statistically significant data about the predictors.

Response

Abstract made more informative by adding the number of patients in both groups and statistics added for predictors of dapt noncompliance.

Question 2

Which therapy as DAPT was indicated? Aspirin and Clopidogrel in all patients? And the length of the therapy was previously established according to some risk score, bleeding risk of the patients or the complexity of the PCI?

Response

Only aspirin and clopidogrel is available through government reimbursement so these two were used in our patient cohort. Length of therapy is determined by the institutional guideline of 12 months of dapt regardless of ACS or CCS and continuing further who tolerate dapt with precice dapt score of less than 25.

Question 3

“DAPT non-adherence was defined as cessation of antiplatelet medication without any side effects or physician order.” What in case of side effect such as thrombosis? How do you consider these patients?

Response

Complication such as stent thrombosis was defined in myocardial infarction in the methods section and considered as such. Those patients who left dapt on physician’s orders or bleeding or other complications were excluded from this study. 

Question 4 and 5

The Target Lesion Revascularization is not considered. Can you explain the reason for this choice? The last sentence of the Statistical analysis paragraph misses the final period.

Response

Target lesion revascularization was considered but it came out to be non-significant among non-compliant individuals and between ad hoc and planned pci. Period added at the end.

Question 6 and 7

Please add the statistical value in the Results for the predictors. Please specify the abbreviations of the statistical value of the Table 4 in its caption (e.g. B, S.E.).

Response

Statistical values added in results for predictors and abbreviations added in legend of table 4.

Question 8

You showed only the predictors with statistically significant value, but you discussed the predictors of the PARIS and EDUCATED trial. What about the impact of the predictors of these trials in your population? Please, it could be relevant if you have the chance to produce the statistical analysis of these predictors and discuss them.

Response

Very avid observation dear sir. We only included the predictors in our cohort that were statistically significant in our population and discussed predictors of PARIS and EDUCATE registry because they differed a bit with our patient cohort. This shows regional variation as dapt and medication compliance as a whole is concerned. That is why we discussed these two registries. 

Regards to the respected editor and reviewers.

---

## [Decision Letter · Decision Letter 1]

7 Jul 2021

Incidence, Predictors, and Outcomes of DAPT Non-compliance in Planned vs. Ad Hoc PCI in Chronic Coronary Syndrome

PONE-D-21-12701R1

Dear Dr. Malik,

We’re pleased to inform you that your manuscript has been judged scientifically suitable for publication and will be formally accepted for publication once it meets all outstanding technical requirements.

Kind regards,

R. Jay Widmer

Academic Editor

PLOS ONE

Additional Editor Comments (optional):

Reviewers' comments:

Reviewer's Responses to Questions

**Comments to the Author**

1. If the authors have adequately addressed your comments raised in a previous round of review and you feel that this manuscript is now acceptable for publication, you may indicate that here to bypass the “Comments to the Author” section, enter your conflict of interest statement in the “Confidential to Editor” section, and submit your "Accept" recommendation.

Reviewer #1: All comments have been addressed

2. Is the manuscript technically sound, and do the data support the conclusions?

Reviewer #1: Yes

3. Has the statistical analysis been performed appropriately and rigorously? 

Reviewer #1: Yes

4. Have the authors made all data underlying the findings in their manuscript fully available?

Reviewer #1: No

5. Is the manuscript presented in an intelligible fashion and written in standard English?

Reviewer #1: Yes

6. Review Comments to the Author

Reviewer #1: Dear Authors,

thank you for your sincere responses.

The clarifications and modifications you applied to the manuscript improved it despite the intrinsic and mainly acknowledged limitations.

Best regards

7. PLOS authors have the option to publish the peer review history of their article (what does this mean?). If published, this will include your full peer review and any attached files.

Reviewer #1: **Yes: **Alessandro Sticchi

---

## [Editor Report · Acceptance letter]

9 Jul 2021

PONE-D-21-12701R1 

Incidence, Predictors, and Outcomes of DAPT Non-compliance in Planned vs. Ad Hoc PCI in Chronic Coronary Syndrome 

Dear Dr. Malik:

I'm pleased to inform you that your manuscript has been deemed suitable for publication in PLOS ONE. Congratulations! Your manuscript is now with our production department. 

Kind regards, 

on behalf of

Dr. R. Jay Widmer 

Academic Editor

PLOS ONE